# Role of cytoneme structures and extracellular vesicles in *Trichomonas vaginalis* parasite-parasite communication

**Nehuén Salas[1,2†], Manuela Blasco Pedreros[1,2†], Tuanne dos Santos Melo[3], Vanina G Maguire[4], Jihui Sha[5], James A Wohlschlegel[5], Antonio Pereira-Neves[3], Natalia de Miguel[1,2\***

[1]Laboratorio de Parásitos Anaerobios, Instituto Tecnológico Chascomús (INTECH), CONICET-UNSAM, Buenos Aires, Argentina; [2]Escuela de Bio y Nanotecnologías (UNSAM), Chascomús, Argentina; [3]Departamento de Microbiologia, Instituto Aggeu Magalhães, Fiocruz, Recife, Pernambuco, Brazil; [4]Área de mejoramiento genético vegetal, Estación Experimental Agropecuaria (EEA INTA), Manfredi, Argentina; [5]Department of Biological Chemistry, University of California, Los Angeles, Los Angeles, United States

**\*For correspondence:**
ndemiguel@intech.gov.ar

[†]These authors contributed equally to this work

**Abstract** *Trichomonas vaginalis,* the etiologic agent of the most common non-viral sexually transmitted infection worldwide. With an estimated annual prevalence of 276 million new cases, mixed infections with different parasite strains are expected. Although it is known that parasites interact with their host to enhance their own survival and transmission, evidence of mixed infections call into question the extent to which unicellular parasites communicate with each other. Here, we demonstrated that different *T. vaginalis* strains can communicate through the formation of cytoneme-like membranous cell connections. We showed that cytonemes formation of an adherent parasite strain (CDC1132) is affected in the presence of a different strain (G3 or B7RC2). Our findings provide evidence that this effect is contact-independent and that extracellular vesicles (EVs) are responsible, at least in part, of the communication among strains. We found that EVs isolated from G3, B7RC2, and CDC1132 strains contain a highly distinct repertoire of proteins, some of them involved in signaling and communication, among other functions. Finally, we showed that parasite adherence to host cells is affected by communication between strains as binding of adherent *T. vaginalis* CDC1132 strain to prostate cells is significantly higher in the presence of G3 or B7RC2 strains. We also observed that a poorly adherent parasite strain (G3) adheres more strongly to prostate cells in the presence of an adherent strain. The study of signaling, sensing, and cell communication in parasitic organisms will enhance our understanding of the basic biological characteristics of parasites, which may have important consequences in pathogenesis.

## Editor's evaluation

This study presents novel information and describes fundamental findings in various fields of biology. It is of interest to a wide audience, including parasitology, cell-to-cell communication, and the roles of extracellular vesicles. The manuscript presents compelling evidence that will contribute to the advancement of our understanding of how parasitic extracellular vesicles (and cytoneme structures) are formed and participate in intra-species communication.

## Introduction

The flagellated protozoan parasite *Trichomonas vaginalis* is the causative agent of trichomoniasis, the most common non-viral sexually transmitted infection worldwide, with an estimated 276 million new cases annually (*WHO, 2018*). Although asymptomatic infection is common, multiple symptoms and pathologies can arise in both men and women, including vaginitis, urethritis, prostatitis, low birth weight infants and preterm delivery, premature rupture of membranes, and infertility (*Fichorova, 2009*; *Swygard et al., 2004*). *T. vaginalis* has also emerged as an important cofactor in amplifying the HIV spread as individuals infected with *T. vaginalis* have a significantly increased incidence of HIV transmission (*McClelland et al., 2007*; *Van Der Pol et al., 2008*). Additionally, *T. vaginalis* infection increases the risk of cervical and aggressive prostate cancer (*Gander et al., 2009*; *Twu et al., 2014*). Due to its great prevalence in some communities, mixed infections with several parasite strains are anticipated. In this sense, an analysis performed of 211 *T. vaginalis* samples isolated in five different continents identified 23 cases of mixed infections (10.9%; *Conrad et al., 2012*). The extent to which parasites communicate with each other in mixed infections has been severely underestimated.

The fundamental ability to sense, process, and respond to extracellular signals is shared by all living forms. However, little is known about the mechanism of sensing and signaling in protozoan parasites compared to what is known in other organisms (*Roditi, 2016*). Although it is widely accepted that pathogens interact with their host to enhance their own survival and transmission, communication between unicellular parasites has been poorly studied (*Roditi, 2016*). Although it was originally believed that single-celled microorganisms do not need to cooperate with other members of their own species, in recent years, it has become clear that microbes are social organisms capable of communicating with one another and engaging in cooperative behavior (*Oberholzer et al., 2010*; *Roditi, 2016*). In this sense, social interactions in a population offer advantages over a unicellular lifestyle, including increased protection from host defenses, access to nutrients, exchange of genetic information, and enhanced ability to colonize, differentiate, and migrate as a group (*Oberholzer et al., 2010*; *Roditi, 2016*).

Cells communicate over short or long distances in different ways. Extracellular vesicles (EVs), soluble secreted factors, membrane protrusions, and direct contact between cells are all different forms of cell communication (*Buszczak et al., 2016*; *Matthews, 2021*; *Regev-Rudzki et al., 2013*). Plasma membrane serves as the primary interface between a cell and its environment, playing an essential role in mediating direct contact, sensing environmental factors, and releasing signaling molecules. Cellular protrusions have emerged as a way for cells to communicate with one another. Among different types of cellular protrusions, filopodia are thin cellular extensions that have been observed in many cell types and have been assigned different roles like cell migration, cell adhesion, force generation, wound healing, environmental sensing, antigen presentation, and neuronal pathfinding (*Roy and Kornberg, 2015*). Although their physical properties vary (2–400 µm in length and 0.1–0.3 µm diameter), all are actin-based, they extend and retract at velocities that have been measured as much as 25 µm/min, and their tips can contact other cells (*Roy and Kornberg, 2015*). Their different shapes and roles are reflected in the many names that have been coined: thin filopodia (*Miller et al., 1995*), thick filopodia (*McClay, 1999*), invadopodia (*Chen, 1989*), telopodes (*Popescu and Faussone-Pellegrini, 2010*), tunneling nanotubes (*Rustom et al., 2004*), and cytonemes (*Ramírez-Weber and Kornberg, 1999*). Specifically, cytonemes are considered as thin specialized filopodia that have been shown to traffic signaling proteins such as morphogens, growth factors, and cell determination factors (*Roy and Kornberg, 2015*). Although cytonemes have similar diameters to conventional filopodia (typically smaller than 200 nm), they have the potential to extend up to ~300 nm from the originating cell body and have been observed in both vertebrate and invertebrate systems (*Kornberg and Roy, 2014*).

Alternatively, EVs are also considered key mediators in intercellular communication in many types of cells. They are a group of heterogeneous particles formed by a lipid bilayer containing proteins and nucleic acids (*Abels and Breakefield, 2016*; *Yáñez-Mó et al., 2015*). As proposed by the International Society for EVs, the term "EVs" is referred to all sub-populations of EVs, so it is recommended to use it collectively and universally (*Théry et al., 2018*). Among the various subtypes of EVs, the particles can be defined according to the mode of biogenesis, size, and function into three major categories: (1) exosomes formed due to plasma membrane invagination into multivesicular bodies with size ranging from 40 to 100 nm; (2) microvesicles (MVs), also called shedding vesicles, microparticles or ectosomes, originated from the budding and extrusion of the plasma membrane, with sizes

between 50 and 1000 nm and an asymmetric structure; and (3) apoptotic bodies, with greater sizes (up to 1000 nm) originated from cells in the process of programmed cell death (*Kalra et al., 2012*). Different to cell membrane protrusions, EVs modulate short- and long-range events, allowing cells to communicate even at long distances. These particles regulate physiological processes such as blood coagulation, cell differentiation, and inflammation, as well as pathological processes caused cancer, neurological, cardiovascular, and infectious diseases (*Kao and Papoutsakis, 2019*; *Raposo and Stoorvogel, 2013*; *Yáñez-Mó et al., 2015*). EVs are relevant for the communication between pathogens and host cells (*Drurey and Maizels, 2021*; *Khosravi et al., 2020*; *Nievas et al., 2020*, p. 2021; *Sabatke et al., 2021*; *Torrecilhas et al., 2020*). Specifically, in *T. vaginalis*, the analysis of EVs, both exosomes and MVs, has become a very exciting field in the study of parasite: host interaction, as it has been shown that the formation of EVs increase in the presence of host cells and modulate parasite adherence (*Nievas et al., 2020*; *Nievas et al., 2018a*; *Olmos-Ortiz et al., 2017*; *Rai and Johnson, 2019*; *Twu et al., 2013*). Although the role of EVs in *T. vaginalis*: host interaction has been deeply analyzed (*Nievas et al., 2020*; *Rada et al., 2022*; *Rai and Johnson, 2019*; *Salas et al., 2021*; *Twu et al., 2013*), the understanding of the role of EVs in communication between different parasite strains and the implications in the infection process is still scarce.

Here, we demonstrated that different *T. vaginalis* strains are capable of communicating with one another through the formation of cytoneme-like membranous cell connections. We observed that cytoneme formation of an adherent parasite strain (CDC1132) is affected in the presence of a different strain (G3 or B7RC2). Furthermore, we have shown that this effect on cytoneme formation is contact-independent and that EVs are responsible, at least in part, of the communication between strains. To explain the differential response in cytoneme formation due to the presence of EVs isolated from different strains, we analyzed the EVs' protein content using mass spectrometry and demonstrated that highly specific protein cargo was detected in EVs isolated from different strains. Finally, we showed that parasite adherence to host cells is affected by this communication as binding of adherent *T. vaginalis* CDC1132 strain to prostate cells is significantly higher in the presence of G3 or B7RC2 strains. Importantly, we observed that in the presence of an adherent strain, a poorly adherent parasite strain (G3) adheres more strongly to prostate cells, suggesting that interaction between isolates with distinct phenotypic characteristics may have significant clinical repercussions. The study of signaling, sensing, and cell communication in parasitic organisms will surely enhance our understanding of the basic biological characteristics of parasites and reveal new potential clinical outcomes.

## Results

### *T. vaginalis* adherent strains form abundant membrane protrusions

Visualization of *T. vaginalis* by fluorescent and live-cell microscopy revealed the presence of filamentous structures extending from the surface of some cells (*Video 1*). Due to their morphological appearance (*Figure 1A*), these structures resembled previously described filopodia and cytonemes (*Kornberg and Roy, 2014*). To determine if these structures were related to pathogenesis, we evaluated the presence of filopodia and cytonemes in strains with different adherence capacities to host cells: two poorly adherent strains (G3 and NYU209) and two highly adherent strains (B7RC2 and CDC1132) using the membrane binding lectin wheat germ agglutinin (WGA). When the number of membrane protrusions was quantified, we observed that the highly adherent strains B7RC2 and CDC1132 have greater number of filopodia and cytonemes compared to poorly adherent strains G3 and NYU209 (*Figure 1A*). In concordance, SEM revealed that multiple filopodia and cytonemes originated from the surface of highly adherent CDC1132 parasites and almost no such protrusions were observed in the poorly adherent G3 strain (*Figure 1B*). These tubular extensions appear to be close ended and can be seen protruding from the region where the flagella

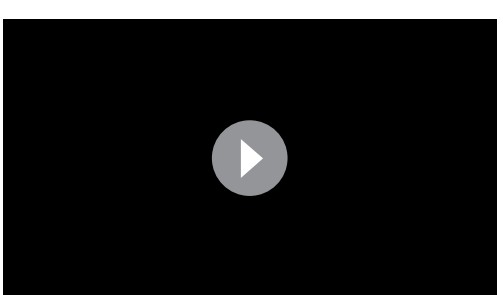

**Video 1.** Live-cell microscopy revealed the presence of filamentous structures extending from the surface of *T. vaginalis*.

https://elifesciences.org/articles/86067/figures#video1

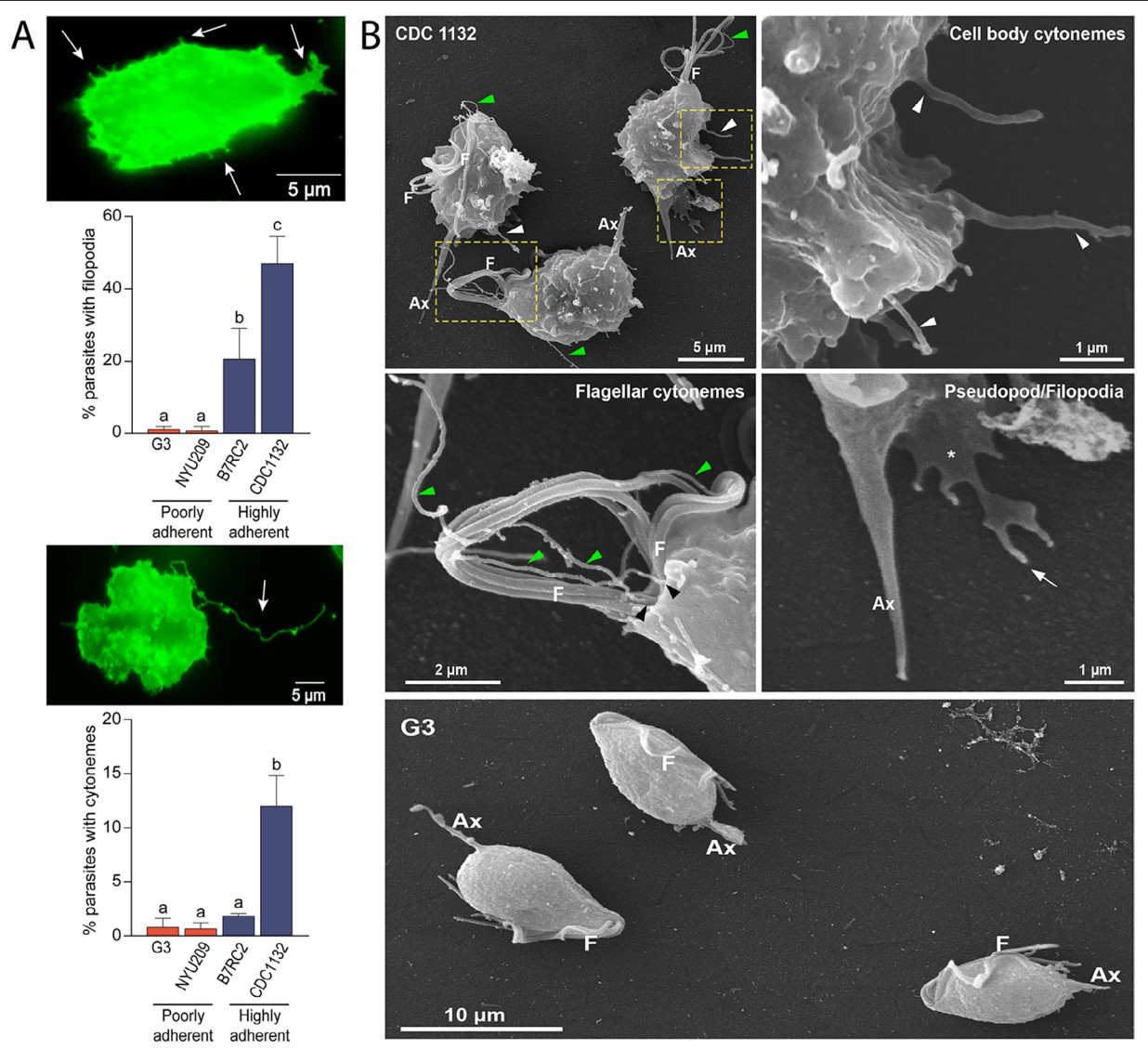

**Figure 1.** Adherent *T. vaginalis* strains form abundant membrane protrusions. (**A**) Quantification of the percentage of parasites containing filopodia (top) or cytonemes (bottom) in their cell surface. The presence of filopodia/cytonemes in two poorly adherent (G3 and NYU209) and two highly adherent strains (B7RC2 and CDC1132) was analyzed. Three independent experiments by duplicate were performed, and 100 parasites were randomly counted per sample. Data are expressed as percentage of parasites with filopodia and cytonemes ± standard deviation. ANOVA followed by Tukey's post hoc test (α=0.95) was used to determine significant differences. (**B**) SEM reveals a myriad of projections originating from the surface of parasites from CDC1132 strain. Cytonemes protruding from cell body and flagellar base region are indicated by white and green arrowheads, respectively. Pseudopodia (*) and filopodia (arrow) are also seen. The surface protuberances appear to be close ended. Almost no protrusions were observed arising from the surface of poorly adherent parasites (G3). F, flagella; Ax, axostyle.

The online version of this article includes the following figure supplement(s) for figure 1:

**Figure supplement 1.** SEM of detailed views of cytonemes protruding from the surface of CDC1132 parasites.

emerge as well as from the cell body, with a length up to 4.6 μm (*Figure 1B*; *Figure 1—figure supplement 1*). The cytonemes from the flagellar base appear unbranched, displaying a homogeneous morphology and size ranging from 70 nm to 100 nm in thickness (*Figure 1B*; *Figure 1—figure supplement 1*). Their ultrastructural features are similar to those flagellar projections previously described in the related veterinary trichomonad *Tritrichomonas foetus* (*Benchimol et al., 2021*). However, the tubular extensions from the cell body are heterogeneous, with diameters that vary from 70 nm to 540 nm and some of them furcated (*Figure 1B*; *Figure 1—figure supplement 1*; *Figure 2—figure supplement 1*). Altogether, these findings demonstrate that *T. vaginalis* contains different types of

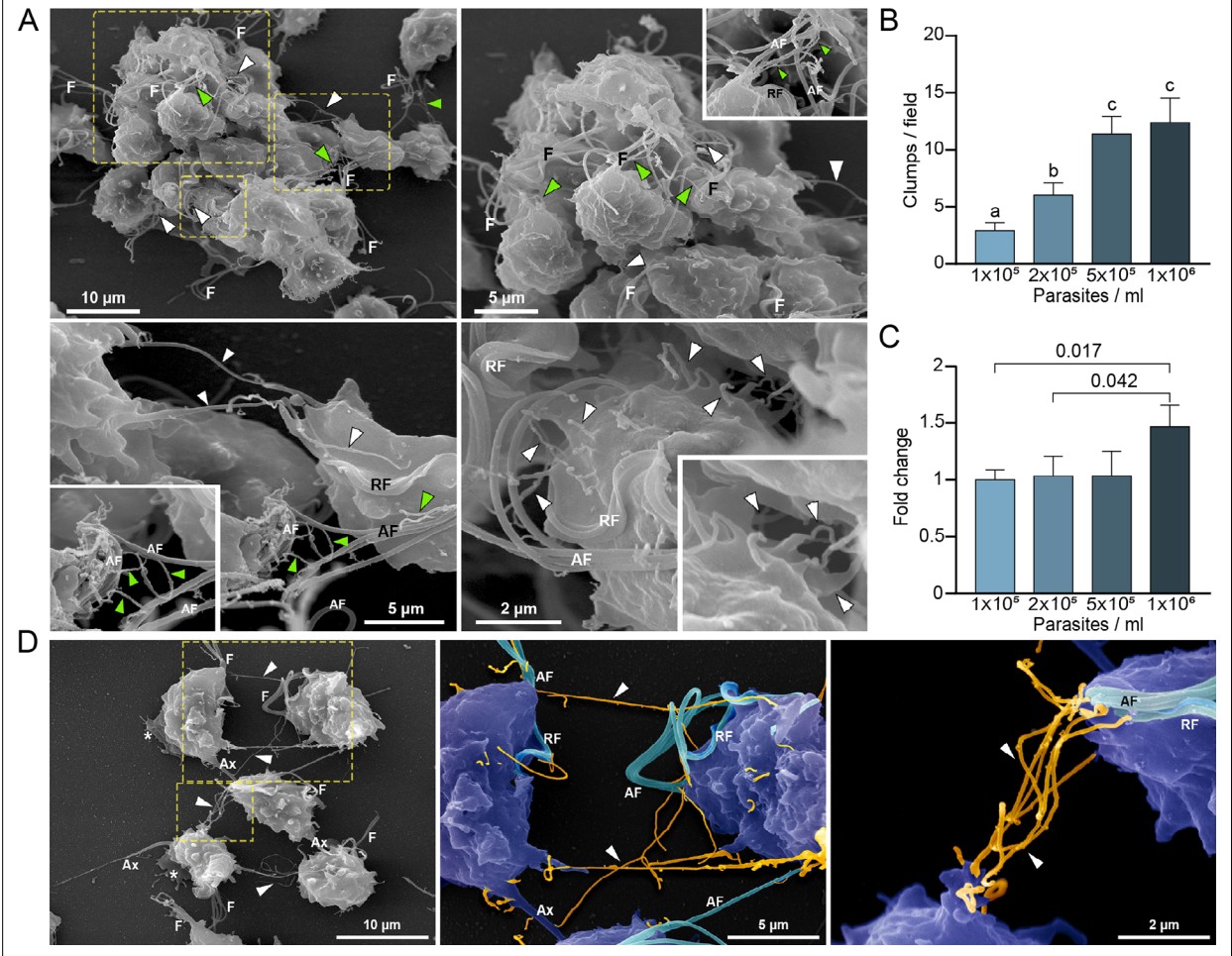

**Figure 2.** Cytonemes are associated to clumps formation. (**A**) Cytonemes emerging from flagella base (green arrowheads) and cell body (white arrowheads) are frequently observed connecting adjacent cells inside clumps of parasites by SEM. F, flagella; AF, anterior flagella; RF, recurrent flagellum. (**B**) Quantification of clumps per field at different parasite densities (parasites/ml). Twenty fields were counted by duplicate in three independent experiments. A clump was defined as an aggregate of ~5 or more parasites. Data are expressed as number of clumps for field ± standard deviation. ANOVA followed by Tukey's post hoc test (α=0.95) was used to determine significant differences. (**C**) Quantification of the number of CDC1132 parasites containing cytonemes at different parasite densities (parasites/ml). Three independent experiments by duplicate were performed, and 100 parasites were randomly counted per sample. Data are expressed as −fold change compared to the number of parasites containing cytonemes at density 1×10$^5$ parasites/ml ± the standard deviation of the mean. Student T-tests (α=0.95) were used to determine significant difference between treatments. (**D**) Cytonemes (orange) are observed connecting two parasites (blue) by SEM (arrowheads). F, flagella; AF, anterior flagella; RF, recurrent flagellum; Ax, axostyle.

The online version of this article includes the following figure supplement(s) for figure 2:

**Figure supplement 1.** SEM of detailed views and insets from *Figure 2D*.

membrane protrusions arising from their surface and suggest that its abundance depends, at least in part, on the phenotype of the strain.

## Cytonemes are associated with parasite clumps formation

As previously shown (*Coceres et al., 2015*; *Lustig et al., 2013*), formation of clumps in cell culture generally correlates with the ability of the strain to adhere and be cytotoxic to host cells. Specifically, highly adherent strains tend to aggregate when cultured in the absence of host cells in contrast to poorly adherent strains that generally do not form clumps in vitro (*Coceres et al., 2015*; *Nievas et al., 2018b*). Based on this observation and our SEM results demonstrating that cytonemes are usually detected connecting different parasites inside the clumps (*Figure 2A*), we evaluated whether the formation of clumps (*Figure 2B*) is accompanied by an increase in the number of parasites containing

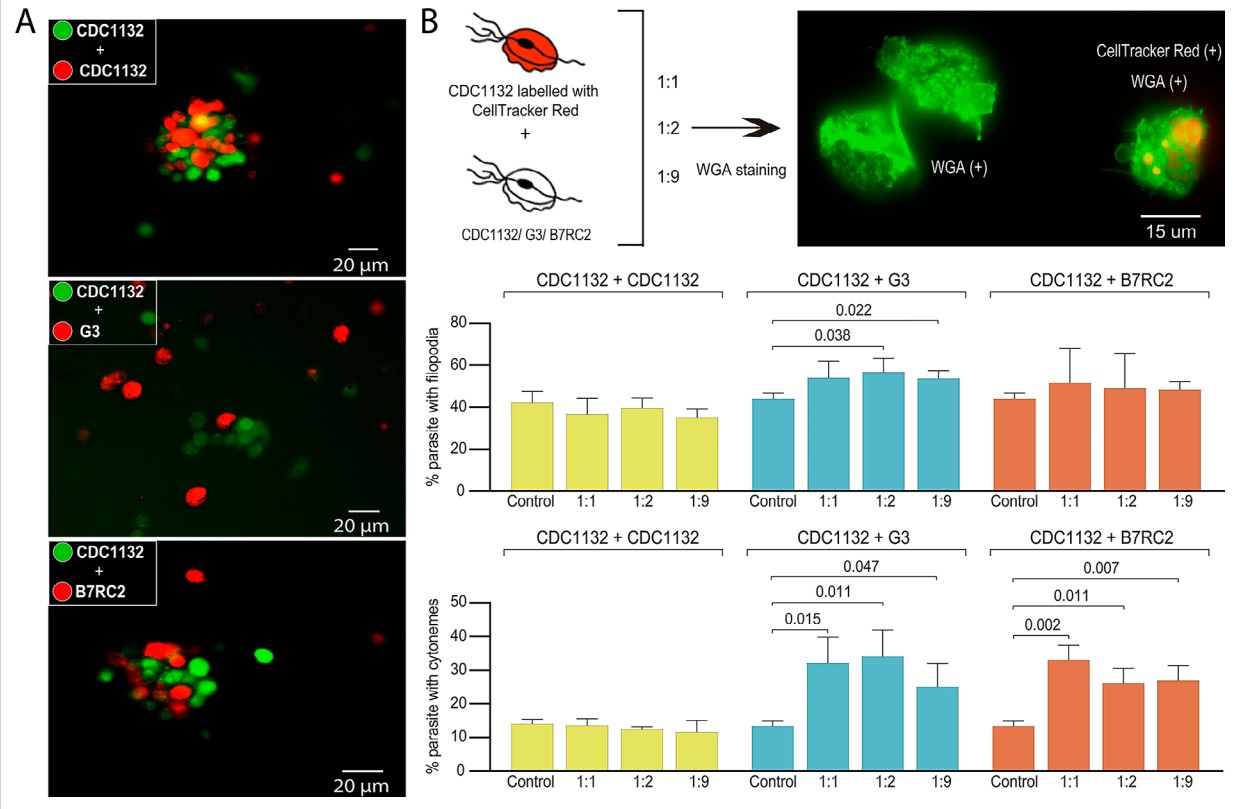

**Figure 3.** Cytoneme formation is induced by interaction between different strains. (**A**) CDC1132 parasites stained with CFSE (green) were co-incubated with CDC1132, G3, or B7RC2 stained with Cell Tracker CMTPX Dye (red) for 1 hr. The interaction between different strains was evaluated by analyzing the capacity to form clumps. As G3 is a poorly adherent strain, the observation of smaller clumps when incubated with CDC1132 is expected. (**B**) Percentage of CDC1132 parasites containing filopodia or cytonemes during co-incubation with different strains. CDC1132 parasites were stained with Cell Tracker Red (red) and co-incubated for 1 hr with different ratio (1:1, 1:2, and 1:9) of unstained CDC1132, G3, and B7RC2. Then, all the parasites were stained with wheat germ agglutinin (WGA; green), and the number of CDC1132 parasites (identified as the parasites stained red and green) containing filopodia (middle panel) and cytoneme (lower panel) were analyzed. As control, the number of cytonemes and filopodia of CDC1132 parasites (without co-incubation) was quantified. Data are expressed as percentage of CDC1132 parasites with filopodia and cytonemes ± standard deviation of three independent experiments. Student T-tests (α=0.95) were used to determine significant difference between treatments.

cytonemes (*Figure 2C*). As can be observed in *Figure 2*, the number of parasites containing cytonemes is higher at $10^6$ parasites/ml (*Figure 2C*); a cell culture condition where usually clumps are formed (*Figure 2B*). These results suggest that cytonemes might be involved in parasite: parasite communication. Consistent with this, cytonemes are frequently observed connecting two parasites (*Figure 2D*). The closed tip of cytonemes was frequently seen in contact with any region of the surface of an adjacent parasite (*Figure 2—figure supplement 1*), including other cytonemes (*Figure 2A and D*).

## Interaction between different strains induce formation of membrane protrusions

Due to the high prevalence of *T. vaginalis*, mixed infections with different parasite strains have been observed (*Conrad et al., 2012*). However, the extent to which parasites communicate and interact with each other during infection has never been analyzed. To evaluate if different parasite strains are able to interact, CDC1132 parasites stained with CFSE(Carboxyfluorescein succinimidyl ester) (green) were co-incubated for 1 hr with a highly adherent strain (B7RC2) or a poorly adherent strain (G3) stained with CellTracker CMTPX Dye (red). As control, CDC1132 stained with CFSE was also co-incubated with CDC1132 stained with CellTracker CMTPX Dye (red). As observed in *Figure 3A*, CDC1132 parasites are able to interact and form clumps with parasites from B7RC2 or G3 strains (*Figure 3A*). As G3 is a poorly adherent strain and usually do not form clumps (*Coceres et al., 2015*), the formation of smaller clumps when incubated with CDC1132 is expected (*Figure 3A*). On the basis of these

findings and the observation that cytonemes are usually found connecting parasites (*Figure 2D*), we hypothesized that communication among different *T. vaginalis* strains might involve the formation of filopodia and/or cytonemes. To evaluate this, CDC1132 strain was stained with CellTracker CMTPX Dye (red) and co-incubated with varying amount of unstained G3, B7RC2, and CDC1132 strains (ratios of 1:1; 1:2; 1:9). Then, co-incubated parasites were stained with WGA, and the formation of filopodia and cytonemes in CDC1132 strain was evaluated by fluorescent microscopy (CDC1132 was visualized as the cells stained with red and green). Although we could not observe a clear increase in filopodia formation in CDC1132 strain upon exposure to G3 or B7RC2, the formation of cytonemes in CDC1132 is clearly affected by the presence of a different strain compared to the exposure to CDC1132 itself (*Figure 3B*). These results suggest that cytonemes are specifically involved in parasite: parasite communication.

## Strain specific parasite-secreted EVs promote cytoneme formation

To evaluate if the observed increase in cytoneme formation in CDC1132 strain due to the presence of a different strain is the result of physical cell contact, we exposed the CDC1132 strain to G3, B7RC2, and CDC1132 (control) strains using a 1 µm porous membrane that prevent direct contact between parasites. This system allows for secreted factors and small EVs to pass between the two cell populations but keeps the parasites (10–15 µm in diameter) loaded in the inserts from contacting the parasites inoculated in the bottom. To measure paracrine signal activation, the number of CDC1132 parasites in the bottom of the well containing cytonemes and filopodia was measured by WGA staining after 1 hr incubation (*Figure 4A*). As can be observed in *Figure 4A*, the number of receiving parasites containing cytonemes and filopodia significantly increases when G3 or B7RC2 strains are loaded in the inserts, compared to the control where CDC1132 strain was loaded in the chamber. These results demonstrated that the effect in cytoneme formation is, at least partially, contact independent. Additionally, these results also suggest that cytoneme formation is affected by paracrine communication among *T. vaginalis* strains.

As the cell culture inserts system allows for secreted factors as well as EVs to pass throughout the filter, we evaluated whether EVs have a role in cytoneme and filopodia formation of recipient parasites. To this end, we conducted a transwell assays using parasites that overexpress the VPS32 protein, a member of the ESCRT III complex, which has been demonstrated to regulate EVs biogenesis and protein cargo sorting in the parasite *T. vaginalis* (*Salas et al., 2021*). As shown in *Figure 4B*, the number of CDC1132 parasites in the bottom of the well containing cytonemes and filopodia is significantly higher when VPS32 overexpressing parasites (G3 strain) are loaded into the inserts compared to the incubation with G3 parasites transfected with empty vector (EpNEO - control parasites) or G3 wild-type parasites (*Figure 4B*). As VPS32-overexpressing produce more EVs than EpNEO parasites, these results may indicate that released EVs have a role in the formation of membrane protrusions in recipient parasites. In order to confirm these results, we isolated EVs from G3, B7RC2, and CDC1132 strains using a previously established protocol (*Salas et al., 2021*), and enriched EVs samples were incubated directly with CDC1132 parasites (*Figure 4C*). The addition of 10 µg of EVs from G3 or B7RC2, but not EVs from CDC1132, increased the formation of cytonemes and filopodia of CDC1132 recipient parasites (*Figure 4C*). As a control, CDC1132 parasites were incubated with PBS solution as well as EVs inactivated by autoclaving (*Schulz et al., 2020*; *Figure 4C*). Similar as previously described where the maximum effect of EVs in *T. vaginalis* was observed at 9 µg (*Twu et al., 2013*), we did not observe any additional increase in the formation of cytonemes and filopodia when increased the concentration of EVs to 20 µg (*Figure 4—figure supplement 1*). These results confirm that EVs are key factors for communication between different strains and suggest that specific molecules are loaded in EVs isolated from different strains.

## Highly specific protein cargo was detected in EVs isolated from different strains

To explain the differential response in cytoneme and filopodia formation induced by the presence of EVs isolated from different strains, we analyzed the protein content of the EVs using mass spectrometry. Three individual samples of EVs isolated from G3, B7RC2, and CDC1132 parasite strains were analyzed (*Figure 5A*). Principal-component analysis confirmed that the signatures of EVs enriched samples isolated from the different strains were clearly distinct (*Figure 5A*). Overall, we

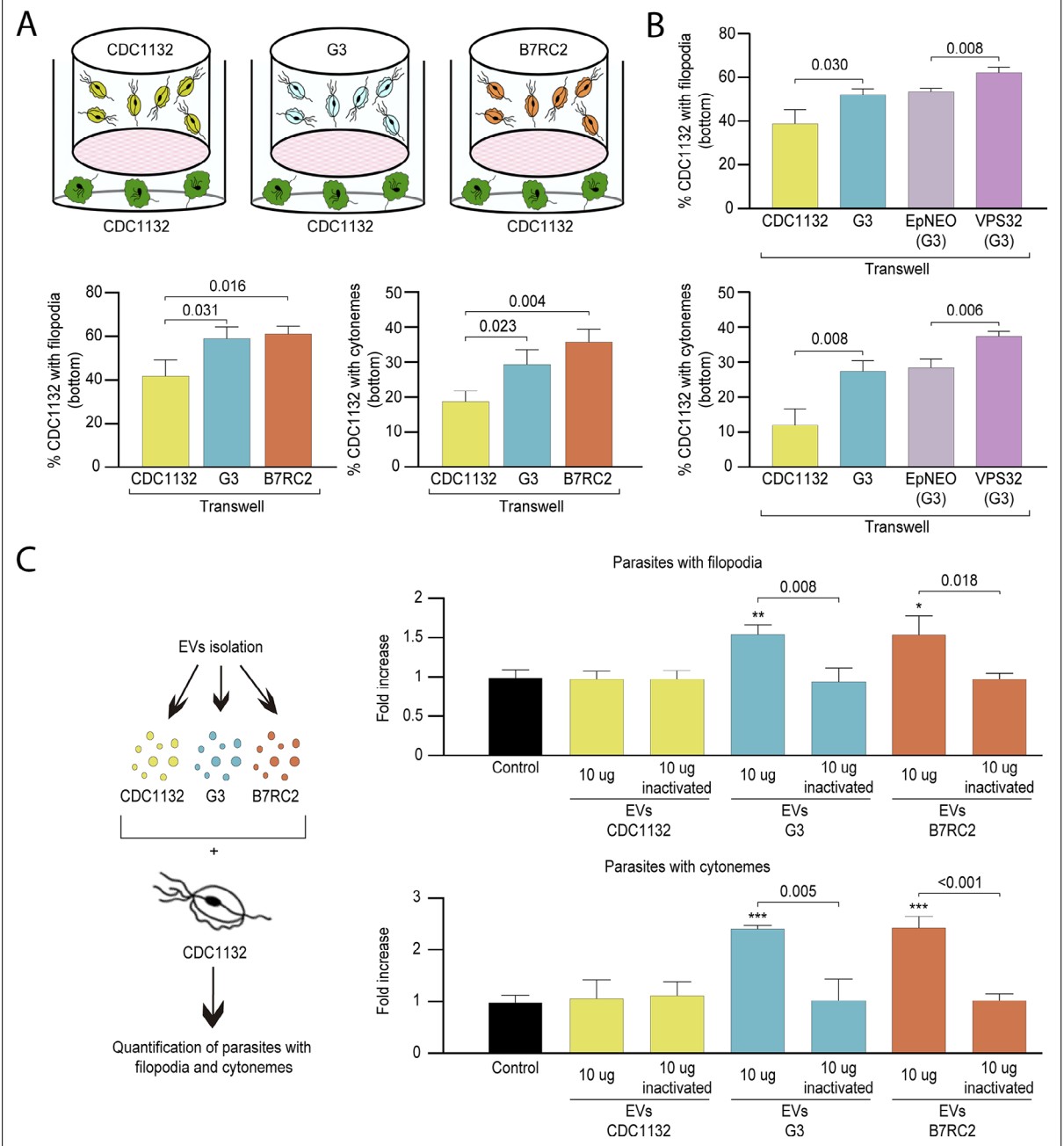

**Figure 4.** Cytoneme formation is induced by paracrine communication. (**A**) Using a cell culture insert assay, CDC1132 parasites (bottom) were co-cultured with CDC1132, G3, and B7RC2 strains (transwell) for 1 hr. Then, the number of CDC1132 parasites (bottom) containing filopodia and cytonemes was quantified by wheat germ agglutinin (WGA) staining. Data are expressed as percentage of parasites with filopodia and cytonemes ± standard deviation of three independent experiments. Student T-tests (α=0.95) were used to determine significant difference between treatments. (**B**) Using a cell culture insert assay, CDC1132 parasites (bottom) were co-cultured with CDC1132, G3, G3 transfected with an empty plasmid (EpNEO) and G3 transfected with VPS32 (transwell) for 1 hr. Then, the number of CDC1132 parasites (bottom) containing filopodia and cytonemes was quantified by WGA staining. Data are expressed as percentage of parasites with filopodia and cytonemes ± standard deviation of three independent experiments. Student T-tests (α=0.95) were used to determine significant difference between treatments. (**C**) Extracellular vesicles (EVs; 10 µg) isolated from G3, B7RC2, and CDC1132 parasites were incubated with wild-type CDC1132 parasites for 1 hr. As control, CDC1132 parasites were incubated with the same volume of the PBS solution or 10 µg of inactivated EVs from different strains. Then, the parasites were stained with WGA, and the number of parasites containing cytonemes and filopodia was quantified. Data are expressed as a mean −fold increase compared to control (without EVs) ± standard deviation of three independent experiments. Student T-tests (α=0.95) were used to determine significant difference between treatments (*p<0.05, **p<0.01, and ***p<0.001).

The online version of this article includes the following figure supplement(s) for figure 4:

*Figure 4 continued on next page*

*Figure 4 continued*

**Figure supplement 1.** Effect of CDC1132 incubation with different concentrations of extracellular vesicles (EVs).

identified 1317, 954, and 375 proteins in the EVs enriched samples isolated from G3, CDC1132, and B7RC2 strains, respectively (*Figure 5B* and *Supplementary file 1*). Although EVs have been isolated from the same number of parasites, the number of proteins identified in the proteome from G3 and CDC1132 was consistently higher than the number of proteins present in B7RC2 samples (*Figure 5B*). Comparison of proteins detected in the EVs proteomes from G3, B7RC2, and CDC1132 with previously published *T. vaginalis* secretome (*Štáfková et al., 2018*), sEVs(small extracellular vesicles) (*Govender et al., 2020*; *Rada et al., 2022*; *Twu et al., 2013*), EVs (*Salas et al., 2021*), MVs (*Nievas et al., 2018a*), and/or surface proteome (*de Miguel et al., 2010*) showed significant overlap as 89% of the proteins identified were previously detected in other proteomes (*Supplementary file 1*). When the protein content was analyzed, the results indicated that only a core of 346 proteins is shared among EVs isolated from different strains (*Figure 5B*) as only. Similarly, the dendrogram that resulted from a hierarchical clustering analysis of proteins using Pearson correlation as a distance metric demonstrated that the proteins of EVs samples from the different strains were distinctly different from each other (*Figure 5C*). As expected, genes involved in several biological processes were highly enriched using the gene ontology (GO) analysis (*Supplementary file 1* and *Figure 5D*). Even while the identity of proteins detected in the EVs of each strain differs from one another, the GO analysis indicates that the biological processes detected in the different EVs preparations are highly conserved (*Figure 5D*). Specifically, proteins associated to cellular and metabolic processes, response to stimulus, signaling, developmental process, and locomotion have been detected in similar proportions among the total proteins detected in EV-enriched samples isolated from each of the analyzed strains (*Figure 5D*). Remarkably, EVs derived from parasites from G3, CDC1132, and B7RC2 strains contain several proteins that play crucial roles in regulating the formation of filopodia and/or cytonemes in other cells. These proteins encompass Rho proteins

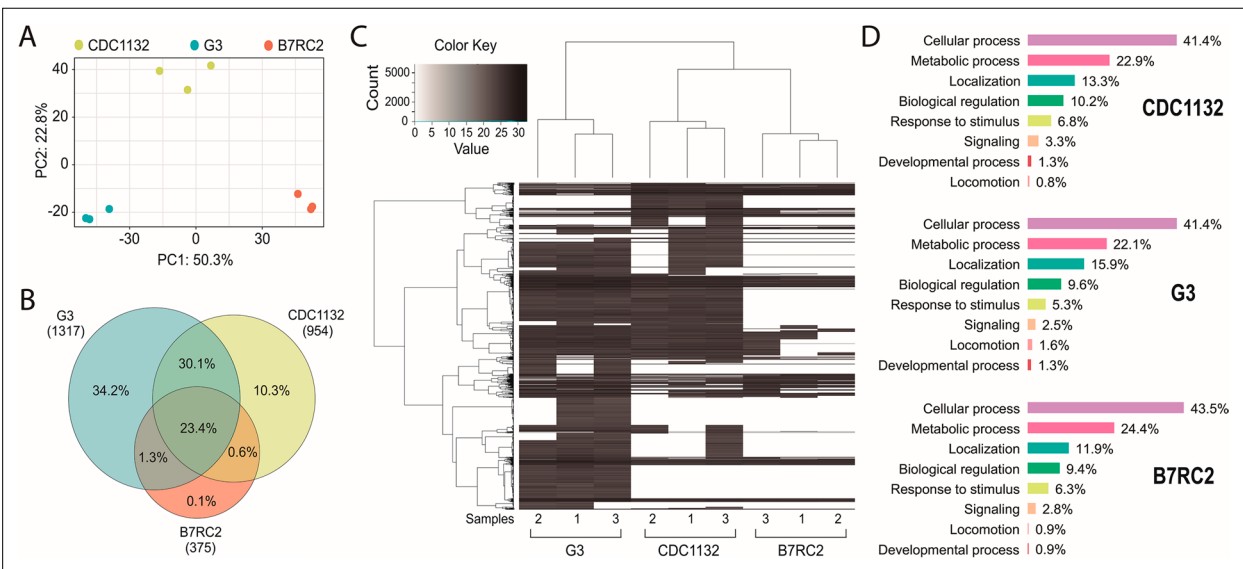

**Figure 5.** Extracellular vesicles (EVs) isolated from different strains contain different protein cargo. EVs' proteomic analysis. (**A**) Principal-component analysis plot representing proteomics data from the comparative analysis of three independent samples of EVs isolated from CDC1132, G3, and B7RC2 strains. (**B**) Venn diagram depicting proteins shared among EVs isolated from G3, B7RC2, and CDC1132 strains. The numbers in parenthesis indicate the quantity of proteins identified in EVs isolated from each strain. The percentages in the diagrams represent the proteins shared by the different vesicles populations considering the total number of proteins identified in the three strains (1480 proteins). (**C**) Heatmap of proteins present EVs of G3, B7RC2, and CDC1132 strains. Three independent isolated samples of EVs isolates from each strain were analyzed (1, 2, and 3). Each horizontal line representing an individual protein. Color gradient represents the protein abundance. Note that a dendrogram resulted from a hierarchical clustering analysis of proteins using Pearson correlation as a distance metric indicates that the proteins of EVs from different strains were distinct from each other. (**D**) Proteins identified in the proteome of EVs from G3, B7RC2, and CDC1132 were identified using Basic Local Alignment Search Tool (BLAST) analysis and sorted into functional groups based on gene ontology biological processes.

(12, 11, and 5 members in G3, CDC1132, and B7RC2, respectively), Ras family proteins (13, 9, and 3 members in G3, CDC1132, and B7RC2, respectively), Calreticulin (1, 2, and 1 member in G3, CDC1132, and B7RC2, respectively), and profilin (5, 5, and 4 members in G3, CDC1132, and B7RC2, respectively), as detailed in *Supplementary file 1*. Of particular interest, the EVs isolated from all three strains also contained several proteins that could potentially be involved in EV biogenesis, including members of the Clathrin (6, 4, and 2 members in G3, CDC1132, and B7RC2, respectively), Tetraspanin (6, 4, and 0 members in G3, CDC1132, and B7RC2, respectively), and SNARE protein families (13, 10, and 2 members in G3, CDC1132, and B7RC2, respectively; *Supplementary file 1*). These data suggest that EVs may be selectively loaded with specific groups of proteins that may be thus exerting specific functions, serving as part of a strain-specific sorting pathway for delivery of biologically active molecules to neighboring cells.

## Communication between different *T. vaginalis* strains affect parasite adherence to prostate BPH-1 cells

For extracellular pathogens such as *T. vaginalis*, the ability of a parasite to adhere to the host is likely a determinant of pathogenesis. To attach, *T. vaginalis* changes morphology within minutes: the flagellated free-swimming cell converts into the amoeboid-adherent stage (*Kusdian et al., 2013*). As it is well established the connections between pathogenesis and ameboid morphological transitions in *T. vaginalis* (*Kusdian et al., 2013*) and our results here, we investigated whether the presence of a different parasite strain could affect the ability of the recipient parasite to convert to an ameboid form (*Figure 6A*). Using a cell culture insert assay, we observed that the percentage of amoeboid CDC1132 cells is higher when G3 or B7RC2 are loaded on the inserts compared to the presence of CDC1132 in the chamber (*Figure 6A*). These results indicate that communication among different *T. vaginalis* strain induce ameboid transformation in recipient parasites.

As *T. vaginalis* amoeboid forms have been previously associated to parasite adherence (*Arroyo et al., 1993*), we then evaluated if communication among strains could affect the adherence of CDC1132 to BPH1 prostate cells. To this end, we performed in vitro binding to prostate BPH1 cells experiments using a membrane cell culture inserts system that allow co-incubation of different *T. vaginalis* strains (*Figure 6B*). Results demonstrate that binding of *T. vaginalis* CDC1132 strain to BPH1 is significantly higher when G3 or B7RC2 are loaded on the insert as opposed to the presence of CDC1132 in the chamber (*Figure 6B*). These results suggest that communication among different *T. vaginalis* strain might affect the behavior of recipient parasites. In concordance with our previous results demonstrating that parasite communication induces cytoneme formation (*Figure 3*) and increase in host-cell adherence (*Figure 6B*), we frequently observed cytonemes protruding from both flagellar base and cell body of CDC1132 in contact with BPH1 cells by SEM (*Figure 6C* and *Video 2*). Thin extensions branching from the cytoneme were also seen in close contact with the BPH1 cells (*Figure 6C*).

To test whether the parasite: parasite communication could affect the behavior of a poorly adherent strain, we expanded our attachment assay and examined whether communication between a highly adherent *T. vaginalis* strain (CDC1132) and a poorly adherent strain (G3) could affect the attachment of G3 to BPH1 cells. Importantly, co-incubation of CDC1132 strain with G3 parasites resulted in a twofold increase in G3 attachment to BPH1 (*Figure 6D*). In concordance with previous reports (*Twu et al., 2013*), these data indicate that conditions media from a highly adherent strain can increase parasite attachment to host cells of a less adherent strain. It has been reported that *T. vaginalis* establishes an endosymbiotic relationship with some *Mycoplasma* species (*Dessì et al., 2019*; *Fichorova et al., 2017*; *Margarita et al., 2020*) and that the presence of the bacteria can alter the parasite's ability to adhere to human epithelial cells (*Margarita et al., 2022*). To rule out the possibility that any observed effects were caused by the presence of *Mycoplasma* in the conditioned media rather than by the EVs, the supernatants of all three strains was PCR amplified using the TransDetect PCR *Mycoplasma* Detection Kit (TransGen). As can be observed in *Figure 6—figure supplement 1*, *Mycoplasma* was not detected in the supernatant of any of the analyzed strains. The findings indicate that EVs released by isolates with distinct phenotypic characteristics can affect the behavior of recipient parasite, which could have significant clinical implications in the context of mixed infections.

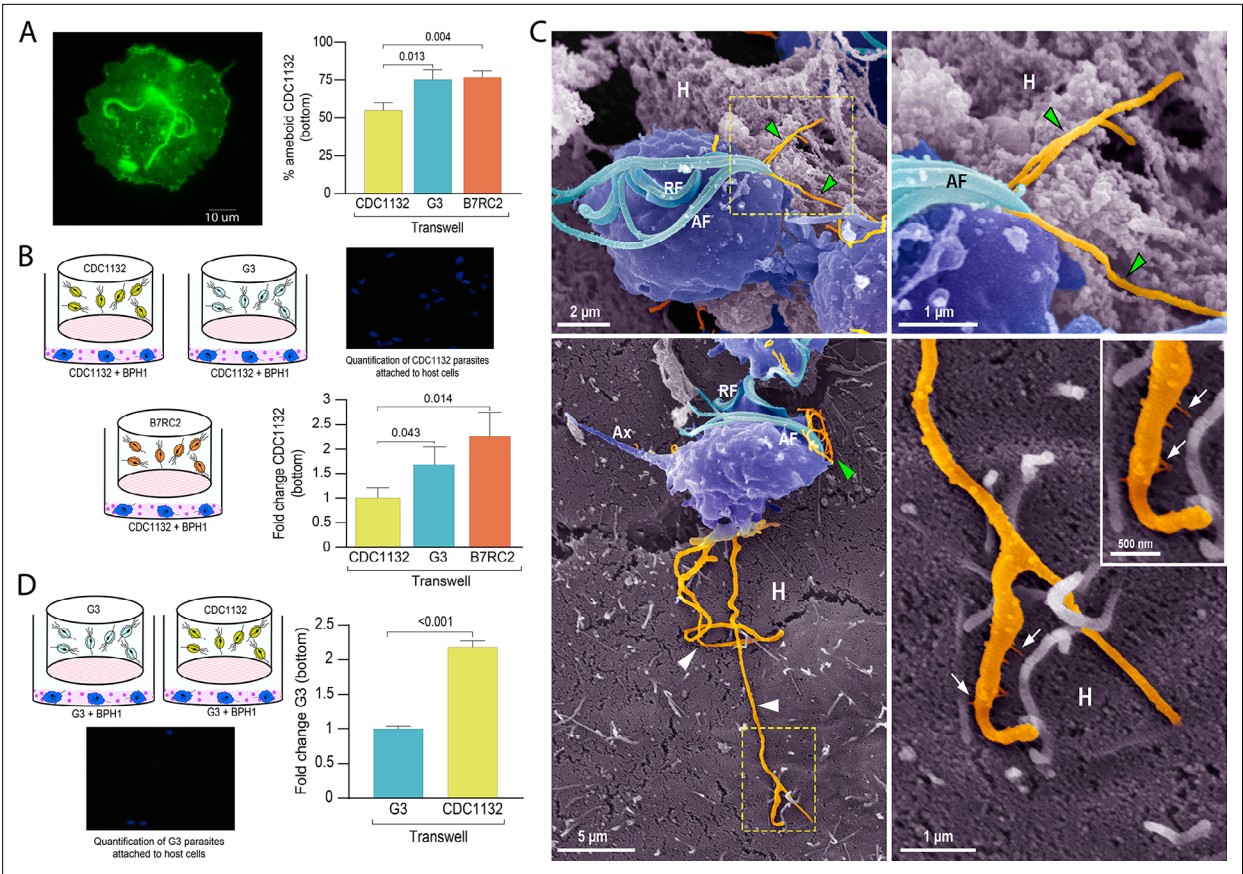

**Figure 6.** Communication between different parasite strains affects attachment to the host cell. (**A**) Using a cell culture insert assay, CDC1132 parasites (bottom) were co-cultured with CDC1132, G3, and B7RC2 strains (transwell) for 1 hr. Then, the percentage of amoeboid CDC1132 parasites in the bottom was quantified by wheat germ agglutinin (WGA) staining. Data are expressed as percentage of ameboid parasites ± standard deviation of three independent experiments. Student T-tests (α=0.95) were used to determine significant difference between treatments. (**B**) Cell Tracker Blue CMAC labeled CDC1132 parasites were incubated for 60 min at 37°C with NhPRE1 prostate cell monolayers cultured onto coverslips in 24-well plates, accompanied by co-culture with CDC1132, G3, and B7RC2 utilizing a cell culture insert assay (1:2 bottom: transwell parasite ratio). Coverslips were washed to remove non-attached parasites and mounted, and the number of attached parasites was quantified by fluorescence microscopy. Data are expressed as fold change related to the attachment of CDC1132 parasites co-incubated with CDC1132 strain inside the transwell ± standard deviation of three independent experiments. Student T-tests (α=0.95) were used to determine significant difference between treatments. (**C**) Cytonemes (orange) protruding from the flagellar base (green arrowheads) and cell body (white arrowheads) of CDC1132 parasites (blue) are observed in contact with host cells (**H**) by SEM. In the lower panels, thin extensions branching (arrows) from the cytoneme are seen in close contact with the BPH1 cells. AF, anterior flagella; RF, recurrent flagellum; Ax, axostyle. (**D**) Cell Tracker Blue CMAC labeled G3 parasites were incubated for 60 min at 37°C with NhPRE1 prostate cell monolayers cultured on coverslips in 24-well plates, accompanied by co-culture with G3 and CDC1132 utilizing a cell culture insert assay (1:2 bottom: transwell parasite ratio). Coverslips were washed to remove non-attached parasites, mounted, and quantified by fluorescence microscopy. Data are expressed as fold change related to the attachment of G3 parasites co-incubated with G3 strain in the transwell ± standard deviation of three independent experiments. Student T-tests (α=0.95) were used to determine significant difference between treatments.

The online version of this article includes the following figure supplement(s) for figure 6:

**Figure supplement 1.** PCR gel electrophoresis results for detection of *Mycoplasma* contamination.

## Discussion

All living organisms, including pathogens, sense extracellular signals and communicate with other cells. Parasites are social organisms capable of communicating with other cells at some stage of their lives to establish and maintain infection (*Lopez et al., 2011*; *Oberholzer et al., 2010*). Most of the research done in this field has specifically focused on analyzing the communication of the parasites with their host, and only a limited number of studies analyzed the communication between parasites. While protozoan parasites typically consider them as individual cells in suspension cultures or animal models of infection, it has been shown that by communicating and acting as a group, unicellular organisms have advantages over individual cells (*Lopez et al., 2011*; *Oberholzer et al., 2010*).

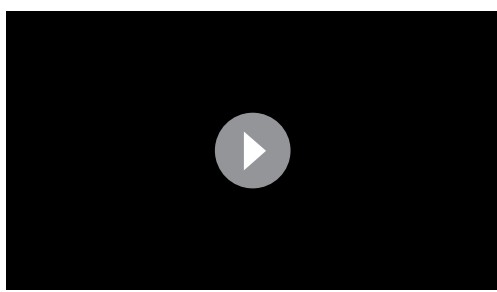

**Video 2.** Cytonemes protruding from CDC1132 parasites in contact with BPH1 cells by videomicroscopy.
https://elifesciences.org/articles/86067/figures#video2

This communication promotes development, survival, access to nutrients and improves the pathogen's defense mechanism against their host (*Ofir-Birin et al., 2017*). As example, communication between *Trypanosoma brucei* parasites modulates the quorum sensing-mediated differentiation of into the tsetse fly transmissible short-stumpy developmental form in the mammalian bloodstream (*Mony et al., 2014*) as well as social motility exhibited by procyclic forms in the insect vector midgut (*Oberholzer et al., 2010*). Research on the malaria parasite *Plasmodium* provides another intriguing example of parasite communication, as studies have revealed that these parasites can alter their sex ratio in response to the presence of unrelated genotypes within the parasite population (*Pollitt et al., 2011*). Although communication among parasites has clear important implications in biology, it has not yet been deeply studied in *T. vaginalis*. Here, we demonstrated that EVs are involved in communication among different *T. vaginalis* parasite strains. Like other eukaryotes, research on EVs in protozoan parasites has grown recently, and the data point to the involvement of protozoan EVs in cell communication (*Szempruch et al., 2016a*; *Wu et al., 2018*). EVs modulate gene expression and affect signaling pathways that might result in developmental changes and modulation of immune response that impact the course of the infection (*Nievas et al., 2020*; *Ofir-Birin et al., 2017*). In this sense, EVs isolated from *Trypanosoma cruzi* trypomastigotes lead to parasite spread and survival (*Trocoli Torrecilhas et al., 2009*). While *T. brucei* EVs promote parasite entrance into host cells (*Atyame Nten et al., 2010*; *Geiger et al., 2010*; *Vartak and Gemeinhart, 2007*), exosomes also have a role in communication between parasites by affecting social motility and migration (*Eliaz et al., 2017*). In *T. vaginalis*, EVs have been found to play a crucial role in the pathogenesis of the parasite, as research has shown that they possess immunomodulatory properties and are capable of modulating the adherence of the parasite to host cells. (*Nievas et al., 2018a*; *Olmos-Ortiz et al., 2017*; *Rai and Johnson, 2019*; *Twu et al., 2013*). While the importance of EVs in facilitating communication between *T. vaginalis* and its host has been established, their role in promoting parasite-to-parasite communication remains relatively unexplored. In this regard, a previous study showed that pre-incubation of small EVs isolated from highly adherent strains increased the attachment of a poorly adherent strain (*Twu et al., 2013*), suggesting that small EVs might be involved in both parasite: parasite and parasite: host communication. Consistent with this, our findings suggest that EVs play a role in mediating communication between distinct strains of the parasite. Specifically, we observed that incubation of EVs-enriched samples from G3 or B7RC2 has an effect in the formation of filopodia and cytonemes of CDC1132 strain. Our prior work has established that the isolation protocol used yields EV-enriched samples (*Salas et al., 2021*), but it is worth noting that co-precipitating proteins may also be present in the samples as a result of inherent limitations in the enrichment techniques employed.

To better understand the molecular pathways involved in the differential response in cytonemes and filopodia formation induced by EVs, the protein content of EVs secreted by different *T. vaginalis* strains was analyzed by mass spectrometry. Surprisingly, the number of proteins detected in the EVs proteome of B7RC2 strain was consistently lower than the number of proteins detected in G3 or CDC1132. Consistent with this, previous studies of sEVs and MVs proteomes performed in B7RC2 strain identified 215 and 592 proteins, respectively (*Nievas et al., 2018b*; *Twu et al., 2013*). However, when sEVs proteome of TV79-49c1 strain was analyzed, the number of proteins detected in these samples was 1633 (*Rada et al., 2022*). These and our data (*Nievas et al., 2018b*; *Rada et al., 2022*; *Twu et al., 2013*) suggest that EVs released from different *T. vaginalis* strains are clearly distinct and may package strain-specific proteins. The striking difference in the number and identities of proteins identified in the different EV proteomes suggests that they may serve as part of a specific sorting pathway for delivering active molecules to neighboring cells. However, despite the variation in the proteins released in the EVs, the biological processes identified through GO analysis were highly conserved. This diversity in protein cargo might explain the differential response observed

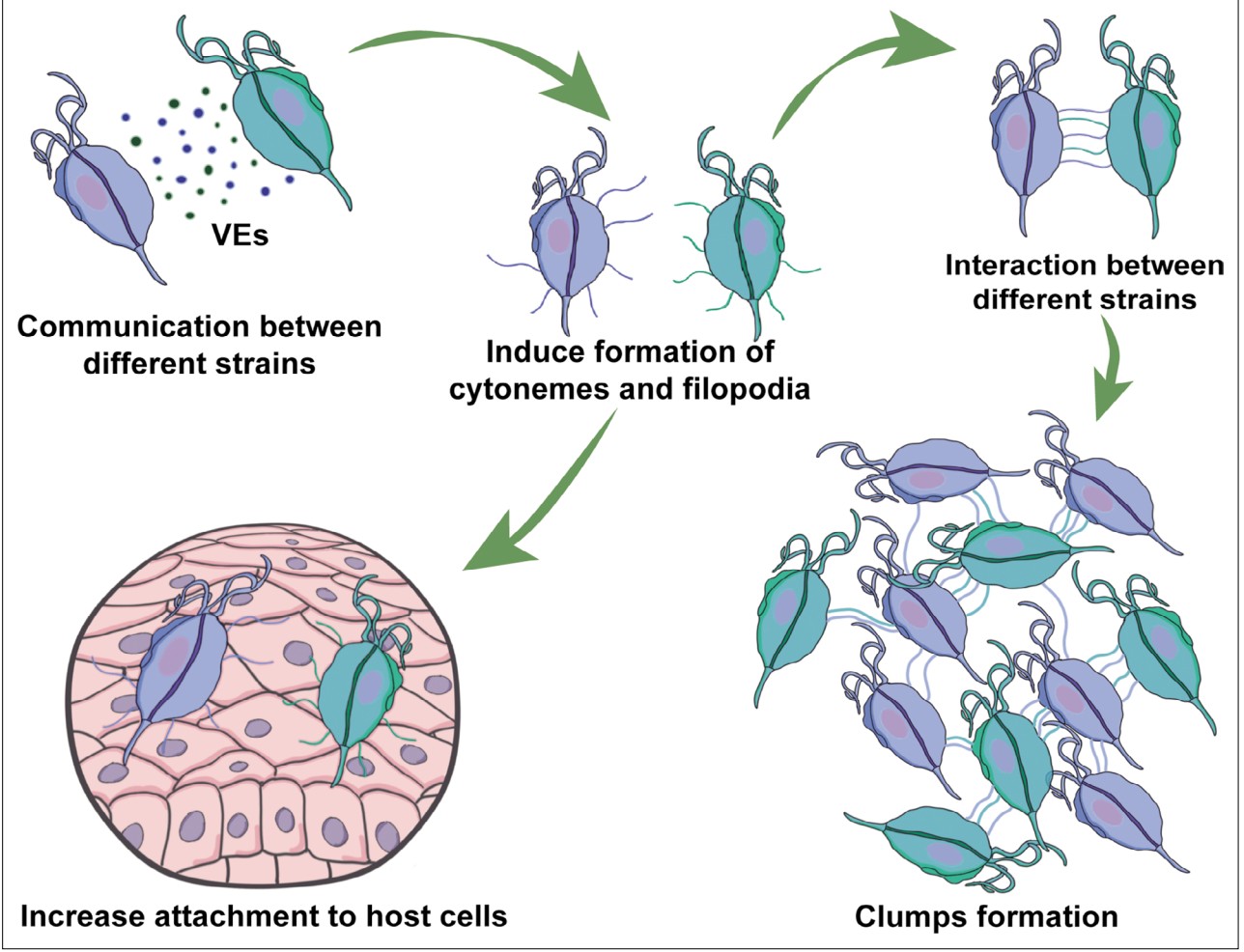

**Figure 7.** Visual summary of role of extracellular vesicles (EVs) and cytonemes in *T.vaginalis* parasite: parasite communication. Different *T. vaginalis* strains release EVs that have specific protein content and affect the formation of filopodia and cytonemes of recipient parasites. The communication among parasites from different strains affects their attachment to host cells, most likely as a result of the increased formation in filopodia and cytonemes structures.

when different EVs are incubated with CDC1132 receiving cells. The data indicates that EVs derived from various strains exhibit a tendency to selectively accumulate particular protein groups, which may consequently fulfill specific functions. Notably, certain proteins that have been identified in the EVs proteomes are linked to the formation of filopodia and/or cytonemes in other cells. For instance, proteins from the Rho and Ras family, which are part of the RAS superfamily of small GTPases that control cell signaling, have been recognized as key regulators in the development of filopodia and cytonemes in other cells (*Koizumi et al., 2012*; *Kozma et al., 1995*; *Pellegrin and Mellor, 2005*). Interestingly, members of these protein families have been detected in the EVs released by all three *T. vaginalis* strains. Likewise, a Calreticulin protein identified in the EVs proteomes of all three strains has been linked to the regulation of cell adhesion, as well as the initiation, stabilization, and turnover of focal contacts (*Opas et al., 1996*; *Villagomez et al., 2009*). EVs isolated from all three strains contain Profilin proteins, which are small actin-binding proteins known to play a critical role in the formation of filopodia (*Rotty et al., 2015*; *Skruber et al., 2020*). Given the significance of these protein families in filopodia and cytonemes formation, it would be valuable to conduct a comprehensive investigation of each of them. Additionally, classical EV proteins, including VPS32 protein (a member of the ESCRT III complex associated with EV biogenesis; *Salas et al., 2021*), proteins involved in membrane trafficking such as Clathrins (*Lafer, 2002*; *Wenzel et al., 2018*), and proteins responsible for vesicle fusion such as SNAREs *Wesolowski and Paumet, 2010*, have also been identified in the EVs proteomes of all three strains analyzed.

Our results demonstrate that communication among strains involve EVs released by different strains that induce the formation of cytonemes and filopodia in recipient cells (*Figure 7*). Cell protrusions are extensions of the plasma membrane of individual cells that function in sensing the environment and making initial dynamic adhesions to extracellular matrix and to other cells (*Adams, 2004*). Specifically, filopodia are thought to transmit signaling molecules to neighboring cells (*Roy et al., 2014*; *Sanders et al., 2013*), increase adhesion (*Albuschies and Vogel, 2013*; *Fierro-González et al., 2013*), and serve as a sensing organelle (*Wood and Martin, 2002*). Cytonemes are specialized types of signaling filopodia that exchange signaling proteins between cells (*Casas-Tintó and Portela, 2019*). Recent research in mammalian cells has proposed filopodia-like cell protrusion as a novel form of intercellular communication. Similarly, bloodstream African trypanosomes also produce membranous protrusions that originate from the flagellar membrane and make connections with the posterior ends of other trypanosomes (*Szempruch et al., 2016b*). The authors also showed that these interactions were stable over long distances (>20 μm) and highly dynamic (*Szempruch et al., 2016b*). A recent discovery has highlighted the crucial role of EVs and cytonemes in intercellular communication. It has been revealed that the cytonemes of cells in the *Drosophila* wing imaginal disc release EVs that carry the hedgehog protein. This protein plays a vital role in cellular signaling and communication, particularly during the development of organs and tissues (*Gradilla et al., 2014*). This discovery highlights the significance of EVs and cytonemes in facilitating cellular communication. In concordance, we describe here that formation of cytonemes in *T. vaginalis* is affected by the presence of EVs from a different strain of the parasite. Additionally, communication among parasites from different strains affect their attachment to host cells, most likely due to increased formation in filopodia and cytonemes structures. Considering that mixed *T. vaginalis* infections have been reported in 10.9% of cases (*Conrad et al., 2012*), our findings may have important clinical implications. We observed that a poorly adherent parasite strain (G3) adheres more strongly to prostate cells in the presence of a highly adherent strain, indicating that the interaction among isolates with distinct phenotypes might affect the behavior of recipient cells, potentially influencing the outcome of infection. Although the relevance of parasite: parasite communication in *T. vaginalis* has not been deeply explored, these results suggest that it could have a significant impact. Furthermore, growing empirical evidence obtained from patients and animal models has shown that multiple-strain infections in different human pathogens can change dynamics, disease course, and transmission (*Balmer and Tanner, 2011*). Multiple-strain infections have been shown unambiguously in 51 human pathogens and are likely to arise in most pathogen species (*Balmer and Tanner, 2011*), indicating that multiple-strain infections are probably the norm and not the exception.

Although the current study did not investigate the genes responsible for membrane structure formation, previous research has revealed that *T. vaginalis* actin and tubulin proteins are known to play a crucial role in the formation of pseudopods, essential cellular structures required for host-cell invasion and mobility (*Lorenzo-Benito et al., 2022*). In concordance, the importance of the actin protein for the flagellated-amoeboid transition had already been demonstrated. By inhibiting actin polymerization, the authors found that the transition did not occur, and the parasite could not migrate through the host tissue (*Kusdian et al., 2013*). Furthermore, amoeboid parasites were able to more easily penetrate the host tissue barrier, and increased tissue invasion was observed with amoeboid parasites (*Kusdian et al., 2013*). Considering the importance of the formation of cytonemes and filopodia in the adherence of parasites to host cells and the known involvement of actin in the formation of these structures (*Casas-Tintó and Portela, 2019*), further investigation into the role of these genes during inter-strain communication is essential. The study of signaling, sensing, and cell communication among parasitic organisms will improve our understanding of host-pathogen interactions and disease dynamics, providing a basis for novel control approaches.

# Materials and methods
## Parasites, cell cultures, and media
*T. vaginalis* strains G3 (ATCC PRA-98; Beckenham, UK), B7RC2 (ATCC 50167; Greenville, NC, USA), and CDC1132 (MSA1132; Mt. Dora, Fla, USA 2008) (*Mercer et al., 2016*) were cultured in TYM (Tryptone-Yeast extract-Maltose) medium supplemented with 10% fetal equine serum and 10 U/ml penicillin (*Clark and Diamond, 2002*). G418 (100 μg/mL; Invitrogen) was added to culture of G3

parasites transfected with VPS32 (TVAG_459530) and EpNEO (control; *Delgadillo et al., 1997*). Parasites were grown at 37°C and passaged daily. The human BPH-1 cells, kindly provided by Dr. Simon Hayward (NorthShore University, USA; *Jiang et al., 2010*), tested for *Mycoplasma* contamination, were grown in RPMI 1640 medium containing 10% fetal bovine serum (Internegocios, Argentina) with 10 U/ml penicillin and cultured at 37°C/5% $CO_2$.

### Detection of *Mycoplasma*

To evaluate the presence of *Mycoplasma* in the supernatant of G3, B7RC2, and CDC1132 cultured parasites, a PCR reaction using TransDetect PCR *Mycoplasma* Detection Kit was performed. To this end, 10 ml of parasite culture with a density of $1.5 \times 10^6$ parasites/ml was centrifuged at 2000× g for 5 min. Then, 40 μl of the cell supernatant was transferred to a PCR tube, incubated at 95°C for 10 min in a thermocycler, and 2 μl was used to proceed with the PCR. Control template provided in the kit and ultrapure water were used as positive and negative control for the reaction, respectively.

### Parasites fluorescent labeling

Parasites were incubated at 37°C for 1 hr on glass coverslips as previously described (*de Miguel et al., 2010*). Parasites attached to coverslips were fixed in 4% paraformaldehyde at room temperature for 20 min and labeled with WGA lectin from *Triticum vulgaris* conjugated with FITC(fluorescein isothiocyanate) (Sigma). To this end, parasites were incubated 1:100 WGA/PBS dilution at 37°C during 1 hr, washed three times with PBS solution and mounted using Fluoromont Aqueous Mounting Medium (Sigma). Fluorescent parasites were visualized using a Zeiss Axio Observer 7 (Zeiss) microscope.

### Scanning electron microscopy

Cells were washed with PBS solution and fixed in 2.5% glutaraldehyde in 0.1 M cacodylate buffer, pH 7.2. The cells were then post-fixed for 15 min in 1% $OsO_4$, dehydrated in ethanol, and critical point-dried with liquid $CO_2$. The dried cells were coated with gold–palladium to a thickness of 15 nm and then observed with a Jeol JSM-5600 SEM, operating at 15 kV. Some images were colored using Adobe Photoshop software (Adobe USA), version 24.0.1.

### Parasite aggregation

Clumps formation was analyzed in parasites grown in regular TYM media under anaerobic conditions at different concentration. A clump was defined as the size corresponding to an aggrupation of at least five parasites. Quantification of clumps in thirty 20× magnification fields was performed using a Nikon TSM (Nikon) microscope. Three independent experiments were performed.

### Direct co-culture of parasites

CDC1132 parasites were labeled using CellTracker Red CMTPX Dye (ThermoFisher). Then, labeled CDC1132 parasites were co-incubated for 1 hr at 37°C with unlabeled G3, B7RC2, and CDC1132 parasites at different cell ratios (1:1, 1:2, and 1:9). Parasites attached to coverslips were fixed and labeled with WGA as described previously. The number of CDC1132 parasites containing filopodia and cytonemes, visualized as the ones with double labeling (red and green), was analyzed using a Zeiss Axio Observer 7 (Zeiss) microscope. Three independent experiments, each of them in duplicates, were performed.

### Indirect parasites co-culture using cell culture inserts assays

Wild-type parasites from G3, B7RC2, and CDC1132 strains as well as G3 parasites transfected with VPS32 or EpNEO were co-cultured using polyester transwell-24 inserts (1 μm pore size; Biofil). CDC1132 parasites were loaded at the bottom of each well and exposed to wild-type G3, B7RC2, CDC1132 parasites, or G3 parasites transfected with VPS32 and EpNEO placed into the transwell inserts (parasite ratio 1:2 bottom: transwell) for 1 hr at 37°C. Then, CDC1132 parasites attached to coverslips were labeled with WGA, and the formation of filopodia and cytonemes was analyzed using a Zeiss Axio Observer 7 (Zeiss) microscope.

### Isolation of *T. vaginalis* EVs

As recommended, the term is referred to all sub-populations of EVs including exosome and MVs (*Théry et al., 2018*). As previously described (*Salas et al., 2021*), EVs were isolated in parallel from

250 ml cultured parasites ($10^6$ cells/ml) from G3, B7RC2, and CDC1132 strains by incubating the parasites for 4 hr at 37°C in TYM medium without serum. Then, conditioned medium was harvested and centrifuged at 3000 rpm for 10 min to remove cell debris. The media was filtered through a 0.8 µm filter, and the sample was pelleted by centrifugation at 100,000× g for 90 min to obtain an EVs enriched fraction (a mixture of MVs and exosomes). The pellet was resuspended in 200 µl cold PBS + 1× cOmplete ULTRA Tablets, Mini, EASYpack Protease Inhibitor Cocktail (Sigma). EVs isolation from G3, B7RC2, and CDC1132 was performed in parallel. Three independent experiments were performed.

## Total protein quantification

Total protein concentration was determined colorimetrically (Bradford Reagent, Sigma-Aldrich). The standard curve was prepared using Bovine Serum Albumin (Promega). Absorbance was measured at 595 nm with a spectrophotometer.

## Incubation of parasites with EVs

To carry out the incubation step, 10 and 20 µg of EVs or the same volume of the PBS solution (control) were incubated for 1 hr at 37°C with CDC1132 parasites in a 24-well plate. As control, EVs (10 µg) were inactivated by autoclaving following a previously described protocol (*Schulz et al., 2020*). After incubation with EVs, parasites were washed twice with PBS and fixed using 4% paraformaldehyde for 20 min. Parasites attached to coverslips were labeled with WGA as previously described, and the formation of cytonemes and filopodia was examined using a Zeiss Axio Observer 7 (Zeiss) inverted fluorescence microscope. Three independent experiments were performed in duplicates.

## Proteomic mass spectrometry analysis

EVs enriched samples were resuspended in a minimal volume of digestion buffer (100 mM Tris–HCl, pH 8, and 8 M urea). Resuspended proteins were reduced and alkylated by the sequential addition of 5 mM tris(2-carboxyethyl) phosphine and 10 mM iodoacetamide as described previously. The samples were then digested by Lys-C (Princeton Separations) and trypsin proteases (Promega; *Florens et al., 2006*). First, Lys-C protease (~1:50 [w/w] ratio of enzyme: substrate) was added to each sample and incubated for 4 hr at 37°C with gentle shaking. The digests were then diluted to 2 M urea by the addition of digestion buffer lacking urea, and trypsin was added to a final enzyme: substrate ratio of 1:20 (w/w) and incubated for 8 hr at 37°C with gentle shaking. Digests were stopped by the addition of formic acid to a final concentration of 5%. Supernatants were carefully removed from the resin and analyzed further by proteomics mass spectrometry. Digested samples were then analyzed using an LC–MS/MS(liquid chromatography-tandem mass spectrometry) platform as described previously (*Kaiser and Wohlschlegel, 2005*; *Wohlschlegel, 2009*). Briefly, digested samples were loaded onto a fused silica capillary column with a 5 µm electrospray tip and packed in house with 18 cm of Luna C18 3 µM particles (Phenomenex). The column was then placed in line with a Q-exactive mass spectrometer (ThermoFisher), and peptides were fractionated using a gradient of increasing acetonitrile. Peptides were eluted directly into the mass spectrometer, where MS/MS spectra were collected. The data-dependent spectral acquisition strategy consisted of a repeating cycle of one full MS spectrum (resolution = 70,000) followed by MS/MS of the 12 most intense precursor ions from the full MS scan (resolution = 17,500; *Kelstrup et al., 2012*). Raw data and spectra analyses were performed using the MaxQuant software (*Tyanova et al., 2016*). For protein identification, a search against a fasta protein database was done consisting of all predicted open reading frames downloaded from TrichDB on November 9, 2022 (*Amos et al., 2022*) concatenated to a decoy database in which the amino acid sequence of each entry was reversed. The following searching parameters were used: (1) precursor ion tolerance was 20 ppm; (2) fragment ion tolerance was 20 ppm; (3) cysteine carbamidomethylation was considered as a static modification; (4) peptides must be fully tryptic; and (5) no consideration was made for missed cleavages. False positive rates for peptide identifications were estimated using a decoy database approach and then filtered using the DTASelect algorithm (*Cociorva et al., 2007*; *Elias and Gygi, 2007*; *Tabb et al., 2002*). Proteins identified by at least two fully tryptic unique peptides, each with a false positive rate of less than 5%, were considered to be present in the sample. Three different sets of EVs enriched samples isolated from G3, B7RC2, and CDC1132 strains were independently analyzed. Proteins present in the EVs fraction were identified

using Basic Local Alignment Search Tool and classified using the GO term enrichment according to PANTHER Classification System (*Mi et al., 2013*).

## Attachment assay

A modified version of an in vitro assay to quantify the binding of *T. vaginalis* to host cell monolayers was carried out (*Bastida-Corcuera et al., 2005*). Briefly, BPH-1 cells were seeded on coverslips in 24-well plates with RPMI(Roswell Park Memorial Institute 1640 medium) culture medium (Invitrogen) and grown to confluence at 37°C and 5% $CO_2$. Parasites (CDC1132 or G3 strain) were labeled with Cell Tracker Blue CMAC (7-amino-4-chloromethylcoumarin; Invitrogen), added to confluent BPH-1 cells (1:3 parasite: host cell ratio), and exposed to different parasites strains loaded in the inserts in a 1:2 bottom: transwell ratio. Plates were then incubated together at 37°C and 5% $CO_2$ for 60 min. Coverslips were subsequently washed in PBS solution, fixed with 4% paraformaldehyde and mounted on slides with Fluoromont Aqueous Mounting Medium (Sigma). Quantification of fluorescent parasites attached to host cells was measured using a Zeiss Axio Observer 7 microscope. Thirty 10× magnification fields were analyzed per coverslip. All experiments were performed three independent times in duplicates.

## Graphics and statistical analyses

Specific statistical considerations and the tests used are described separately for each subsection. GraphPad Prism for Windows version 8.00 was used for graphics. Data are shown as mean ± SD. Statistical significance was established at $p < 0.05$, and for statistical analyses, the InfoStat software (*Di Rienzo et al., 2011*) version 2020e was used.

## Acknowledgements

We thank our colleagues in the lab for helpful discussions. This research was supported with a Grant from the Agencia Nacional de Promoción Científica y Tecnológica (ANPCyT) Grant BID PICT-2019–01671 (NdM) and Arturo Falaschi ICGEB Fellowship (ICGEB). NdM is a researcher from the National Council of Research (CONICET) and UNSAM. NS and MBP are PhD fellow from CONICET. The funders had no role in study design, data collection and analysis, decision to publish, or preparation of the manuscript.

## Additional information

### Competing interests

Natalia de Miguel: Reviewing editor, *eLife*. The other authors declare that no competing interests exist.

### Funding

| Funder | Grant reference number | Author |
| --- | --- | --- |
| Fondo para la Investigación Científica y Tecnológica | PICT-2019-01671 | Natalia de Miguel |
| Arturo Falaschi ICGEB Fellowship (ICGEB) | | Nehuén Salas |
| National Council of Research (CONICET) and UNSAM | | Natalia de Miguel |

The funders had no role in study design, data collection and interpretation, or the decision to submit the work for publication.

### Author contributions

Nehuén Salas, Conceptualization, Data curation, Formal analysis, Investigation, Methodology, Writing - original draft, Writing - review and editing; Manuela Blasco Pedreros, Data curation, Formal analysis,

Validation, Investigation, Methodology, Writing - review and editing; Tuanne dos Santos Melo, Investigation, Visualization, Methodology, Writing - review and editing; Vanina G Maguire, Data curation, Formal analysis, Writing - review and editing; Jihui Sha, Investigation, Methodology, Writing - review and editing; James A Wohlschlegel, Resources, Investigation, Methodology, Writing - review and editing; Antonio Pereira-Neves, Supervision, Investigation, Visualization, Writing - review and editing; Natalia de Miguel, Conceptualization, Supervision, Funding acquisition, Writing - original draft, Project administration, Writing - review and editing

#### Author ORCIDs
Nehuén Salas  http://orcid.org/0000-0002-8841-2717
Natalia de Miguel  http://orcid.org/0000-0002-3864-0703

#### Decision letter and Author response
Decision letter https://doi.org/10.7554/eLife.86067.sa1
Author response https://doi.org/10.7554/eLife.86067.sa2

## Additional files

#### Supplementary files
• Supplementary file 1. List of proteins identified in isolated extracellular vesicles (EVs) from G3, CDC1132, and B7RC2 strains by mass spectrometry. LFQ(label-free quantitation) intensity: Untargeted label-free quantitation of proteins.

• MDAR checklist

#### Data availability
All data available in the manuscript.

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
