## [Editor Report]

This study presents novel information and describes fundamental findings in various fields of biology. It is of interest to a wide audience, including parasitology, cell-to-cell communication, and the roles of extracellular vesicles. The manuscript presents compelling evidence that will contribute to the advancement of our understanding of how parasitic extracellular vesicles (and cytoneme structures) are formed and participate in intra-species communication.

---

## [Decision Letter]

**Decision letter after peer review:**

Thank you for submitting your article "Role of cytoneme-like structures and extracellular vesicles in Trichomonas vaginalis parasite: parasite communication" for consideration by *eLife*. Your article has been reviewed by 3 peer reviewers, and the evaluation has been overseen by a Reviewing Editor and Dominique Soldati-Favre as the Senior Editor. The following individual involved in the review of your submission has agreed to reveal their identity: PierLuigi Fiori (Reviewer #2).

Essential revisions:

1) Please evaluate the feasibility of the suggestions listed by reviewer 1, since they require substantial experimentation. If there are difficulties to follow these suggestions, please provide an explanation of the reasons why your conclusions can be supported without these new assays;

2) Please explore, on the basis of the current literature, how EV and cytoneme formation are related;

3) Please modify the text in accordance with the comments by the three reviewers.

*Reviewer #1 (Recommendations for the authors):*

The most important points and of greatest concern is in relation to experiments with cellular communication.

Figure 4 is very interesting and has strong evidence of cell communication being a specific strain. depending on extracellular vesicles. Experiments A and B are related but may have similar ways of presenting the results to have better deductions. The effect of cellular communication through Transwell is represented by the percentage of parasites that acquire filopodia or cytonemes. In B the experiment is performed by adding extracellular vesicles purified previously and the formation of filopodias and cytonemas is represented by the increased presence by times x. This does not allow us to easily deduce the direct participation of extracellular vesicles in the effect on A or soluble factors. The experiments should be related to each other, where in B The authors could add the supernatant of the interactions between parasites removed from extracellular vesicles to see if there is a synergistic effect on the formation of filopodios or cytonemas or just the effect is by extracellular vesicles. Another control that could be used is the use of inactivated extracellular vesicles. Another important curiosity is because the effect is not dose-dependent, Was there saturation of the effect? In this way, it would be important for the authors to investigate the minimum concentration capable of inducing a significant formation of filopodias or cytonemas and if it responds to a dose dependence or is just a sign of activation by the recognition of specific components of the vesicles involved.

Another suggestion in the experiment in B is to mark the vesicles with fluorochromes and see what happens to them in the parasite. Are they endocited? Are part of the filopodias or cytonemes? The authors could follow the course of the EVS during the parasite-parasite The authors could follow the course of the EVS during the parasite-parasite communication.

The experiments in Figure 6 on the phenotypic effects of the combination of strains are very interesting. The authors could make a set of experiments with a total conditioned medium, with extracellular vesicles of the interaction, and with conditioned medium removed from extracellular vesicles to determine if the effects are due to extracellular vesicles or soluble factors released.

*Reviewer #2 (Recommendations for the authors):*

The manuscript describes the fundamental role of protozoan-to-protozoan and protozoan-to-host cell communication in modulating the pathogenicity of Trichomonas vaginalis.

In fact, the manuscript has many strengths and only a few weaknesses I would ask the authors to clarify just a few points (see below)

1. Many clinical isolates of T.vaginalis establish symbiosis with some Mycoplasma species particularly (M.hominis and M.girerdii). Have the T.vaginalis strains used for the experiments been screened for the presence of the bacteria? It is known that the reference strain G3 is mycoplasma-free, but it is necessary to know whether the other isolates possess bacterial endosymbionts.

2. The presence of endosymbionts influences the pathophysiology of T.vaginalis. In particular, both the adhesiveness and cytopathogenicity of the protozoan are modulated. These aspects should be presented in the discussion, especially the adhesive capabilities of human cells

3. The manuscript does not report a study on the expression of genes involved with adhesiveness or modulation of cytoskeleton architecture in different experimental conditions. This can be considered a limitation of the work and should be reported in the discussion.

*Reviewer #3 (Recommendations for the authors):*

(1) Page-4 (Introduction): It appears that the literature cited about parasite EVs is not updated. Authors might consider revisiting and citing those EV-related papers that are more relevant to the current study.

(2) Figure 5. Efforts should be undertaken to connect the data from the proteomic analysis with biological processes such as cytoneme formation, EV secretion, and pathogenicity caused by Trichomonas.

(3) The authors have carried out solid experiments that produced very interesting results. Unfortunately, they were not discussed in detail in the Results section. The authors might consider re-writing some sections of the results.

(4) How EV secretion and cytoneme formation are related? Is it possible that exosomal vesicles released by T. vaginalis change the membrane structures that lead to cytoneme formation?

(5) Figure 2. The clumps formations by CDC 1132 are shown. It is necessary to show the clumps formation by poorly adherent cells just for comparison. Also, fold changes of what (panel C)?

(6) Some Figure legends (Figure 2, Figure 3, Figure 5, etc.) are short. They should be elaborated accordingly.

(7) In several places in the text, there are some grammatical mistakes, which require the authors' attention.

---

## [Author Response]

Reviewer #1 (Recommendations for the authors):The most important points and of greatest concern is in relation to experiments with cellular communication.Figure 4 is very interesting and has strong evidence of cell communication being a specific strain. depending on extracellular vesicles. Experiments A and B are related but may have similar ways of presenting the results to have better deductions. The effect of cellular communication through Transwell is represented by the percentage of parasites that acquire filopodia or cytonemes. In B the experiment is performed by adding extracellular vesicles purified previously and the formation of filopodias and cytonemas is represented by the increased presence by times x. This does not allow us to easily deduce the direct participation of extracellular vesicles in the effect on A or soluble factors. The experiments should be related to each other, where in B The authors could add the supernatant of the interactions between parasites removed from extracellular vesicles to see if there is a synergistic effect on the formation of filopodios or cytonemas or just the effect is by extracellular vesicles.

We appreciate the reviewer's comment and share his/her opinion. However, it is not feasible to compare experiment 4A with former experiment 4B to fully comprehend the involvement of EVs in the observed effect due to several methodological limitations that prevent direct comparison. The isolation of EVs in *T. vaginalis* is highly inefficient, requiring 250-500 ml of conditioned media for isolation. Furthermore, the standard protocol calls for the use of 10 µg of isolated EVs for experiments, yet the amount of EVs in the transwell assay remains unknown. These differences in the isolation protocol render it impossible to accurately assess the role of EVs in the transwell experiment when using a 24-well plate (Figure 4A). We propose here that EVs play a crucial role in the communication between parasites, although we concur with the reviewer that there may be other secreted factors involved in this process.

To establish the involvement of EVs in parasite: parasite communication within a comparable experimental setup, we have conducted a transwell assay using parasites that overexpress the VPS32 protein. In a previous publication (Salas, et al. 2022), we have demonstrated that parasites with an overexpression of VPS32, a component of the ESCRT III complex, release a larger quantity of EVs compared to the EpNEO control (parasites transfected with an empty plasmid). As shown in the new Figure 4B, the number of CDC1132 parasites located at the bottom of the well containing cytonemes and filopodia is significantly higher when VPS32 overexpressing parasites (G3 strain) are loaded into the inserts compared to the incubation with G3 parasites transfected with EpNEO or G3 wild type parasites (Figure 4B). Given that VPS32-overexpressing parasites release more EVs than EpNEO parasites, these findings provide compelling evidence that the released EVs play a key role in the formation of membrane protrusions in recipient parasites. Notably, this experimental setup is linked to Figure 4A, enabling the contribution of EVs to be evaluated.

Another control that could be used is the use of inactivated extracellular vesicles.

We thank the reviewer for this excellent suggestion. We have followed his/her suggestion and analyzed the effect using inactivated EVs as control. These results have now been included as Figure 4c.

Another important curiosity is because the effect is not dose-dependent, Was there saturation of the effect? In this way, it would be important for the authors to investigate the minimum concentration capable of inducing a significant formation of filopodias or cytonemas and if it responds to a dose dependence or is just a sign of activation by the recognition of specific components of the vesicles involved.

As previously documented in our publications, the optimal effect of EVs on T. vaginalis adherence was found to be at a dose of 9-10 μg (Twu et al., 2013; Salas et al., 2022). In agreement with this observation, no further enhancement in the formation of cytonemes and filopodia was observed when the concentration of EVs was increased to 20 μg, as illustrated in Supplementary Figure S3. Consequently, all experiments were conducted using 10 μg of EVs.

Another suggestion in the experiment in B is to mark the vesicles with fluorochromes and see what happens to them in the parasite. Are they endocited? Are part of the filopodias or cytonemes? The authors could follow the course of the EVS during the parasite-parasite The authors could follow the course of the EVS during the parasite-parasite communication.

Regrettably, we have not yet delved into the mechanism by which EVs impact cytoneme formation in recipient parasites. We acknowledge that this is a fascinating avenue of analysis, and future research will be dedicated to elucidating this mechanism. Nevertheless, we recognize that this is beyond the scope of the present study.

The experiments in Figure 6 on the phenotypic effects of the combination of strains are very interesting. The authors could make a set of experiments with a total conditioned medium, with extracellular vesicles of the interaction, and with conditioned medium removed from extracellular vesicles to determine if the effects are due to extracellular vesicles or soluble factors released.

This would be a very interesting experiment. However, as previously stated, there are methodological limitations that make this experiment impossible.

Reviewer #2 (Recommendations for the authors):The manuscript describes the fundamental role of protozoan-to-protozoan and protozoan-to-host cell communication in modulating the pathogenicity of Trichomonas vaginalis.In fact, the manuscript has many strengths and only a few weaknesses I would ask the authors to clarify just a few points (see below)1. Many clinical isolates of T.vaginalis establish symbiosis with some Mycoplasma species particularly (M.hominis and M.girerdii). Have the T.vaginalis strains used for the experiments been screened for the presence of the bacteria? It is known that the reference strain G3 is mycoplasma-free, but it is necessary to know whether the other isolates possess bacterial endosymbionts.

We thank the reviewer for this suggestion. To rule out the possibility that any observed effects were caused by the presence of Mycoplasma in the conditioned media rather than by the EVs, the supernatants of all three strains was PCR amplified using the TransDetect PCR Mycoplasma Detection Kit (TransGen). These results have now been included as Sup.

Figure 4

2. The presence of endosymbionts influences the pathophysiology of T.vaginalis. In particular, both the adhesiveness and cytopathogenicity of the protozoan are modulated. These aspects should be presented in the discussion, especially the adhesive capabilities of human cells

Done.

3. The manuscript does not report a study on the expression of genes involved with adhesiveness or modulation of cytoskeleton architecture in different experimental conditions. This can be considered a limitation of the work and should be reported in the discussion.

Done.

Reviewer #3 (Recommendations for the authors):(1) Page-4 (Introduction): It appears that the literature cited about parasite EVs is not updated. Authors might consider revisiting and citing those EV-related papers that are more relevant to the current study.

New references have been included.

(2) Figure 5. Efforts should be undertaken to connect the data from the proteomic analysis with biological processes such as cytoneme formation, EV secretion, and pathogenicity caused by Trichomonas.

Done.

(3) The authors have carried out solid experiments that produced very interesting results. Unfortunately, they were not discussed in detail in the Results section. The authors might consider re-writing some sections of the results.

Done.

(4) How EV secretion and cytoneme formation are related? Is it possible that exosomal vesicles released by T. vaginalis change the membrane structures that lead to cytoneme formation?

Regrettably, we have not yet delved into the mechanism by which EVs impact cytoneme formation in recipient parasites. We acknowledge that this is a fascinating avenue of analysis, and future research will be dedicated to elucidating this mechanism. Nevertheless, we recognize that this is beyond the scope of the present study.

(5) Figure 2. The clumps formations by CDC 1132 are shown. It is necessary to show the clumps formation by poorly adherent cells just for comparison. Also, fold changes of what (panel C)?

Poorly adherent strains usually do not form clumps, they grow separately. Panel C legend was clarified.

(6) Some Figure legends (Figure 2, Figure 3, Figure 5, etc.) are short. They should be elaborated accordingly.

Figure legends have been modified accordingly

(7) In several places in the text, there are some grammatical mistakes, which require the authors' attention.

Done.